# Probiotics Can Cure Oral Aphthous-Like Ulcers in Inflammatory Bowel Disease Patients: A Review of the Literature and a Working Hypothesis

**DOI:** 10.3390/ijms20205026

**Published:** 2019-10-11

**Authors:** Francesco Cappello, Francesca Rappa, Federica Canepa, Francesco Carini, Margherita Mazzola, Giovanni Tomasello, Giuseppe Bonaventura, Giovanna Giuliana, Angelo Leone, Dario Saguto, Federica Scalia, Fabio Bucchieri, Alberto Fucarino, Giuseppina Campisi

**Affiliations:** 1Department of Biomedicine and Neurosciences and Advanced Diagnostics, University of Palermo, 90127 Palermo, Italy; francesco.cappello@unipa.it (F.C.); francyrappa@hotmail.com (F.R.); francesco.carini@unipa.it (F.C.); margheritamazzola@hotmail.it (M.M.); giovanni.tomasello@unipa.it (G.T.); giuseppe.bonaventura@unipa.it (G.B.); angelo.leone@unipa.it (A.L.); dariosaguto@gmail.com (D.S.); scalia.fede@gmail.com (F.S.); fabio.bucchieri@unipa.it (F.B.); 2Euro-Mediterranean Institute of Science and Technology (IEMEST), 90139 Palermo, Italy; 3Department of Surgical, Oncological and Stomatological Disciplines, Sector of Oral Medicine, University of Palermo, 90127 Palermo, Italy; canepafederica86@gmail.com (F.C.); giovanna.giuliana@unipa.it (G.G.); giuseppina.campisi@unipa.it (G.C.)

**Keywords:** microbiota, muco-microbiotic layer, dysbiosis, inflammatory bowel diseases, aphthous-like ulcers, probiotics

## Abstract

Dysbiosis has been associated with the onset of several chronic autoimmune or inflammatory pathologies (e.g., inflammatory bowel diseases—IBD), because of its primary role in the establishment of a chronic inflammatory process leading to tissue damage. Inflammatory bowel diseases can even involve areas far away from the gut, such as the extraintestinal manifestations involving the oral cavity with the onset of aphthous-like ulcers (ALU). Studies carried out on animal models have shown that intestinal dysbiosis may be related to the development of autoimmune diseases, even if the mechanisms involved are not yet well known. The aim of this paper is to verify the hypothesis that in inflammatory bowel diseases patients, aphthous-like ulcers are the result of the concomitance of intestinal dysbiosis and other events, e.g., the microtraumas, occurring in the oral mucosa, and that *ex adiuvantibus* therapy with probiotics can be employed to modify the natural course of the aphthous-like ulcers.

## 1. Introduction

The human gut contains a high concentration of bacteria, collectively called microbiota. There are at least 1000 bacterial species (of which 150 to 500 only in the colon) with a density of 9 × 10^13–14^. Several roles, related to the regulation of the host’s physiology, have been attributed to gut microbiota, including stimulation of the immune system, control of pathogenic bacterial proliferation, production of short-chain fatty acids, and fermentation of amino acids and saccharides [1,2].

A disequilibrium of the intestinal microbiota is called dysbiosis, and it causes an alteration of the intercellular tight junctions, allowing access of pathogens (and their toxins, in particular bacterial lipopolysaccharides) and stimulation of the mucosa-associated lymphatic tissue (MALT) with activation of the inflammatory cascade (leukocytes, cytokines, TNF-α), establishment of a chronic inflammation process (Figure 1) and, consequently, tissue damage [1].

For this reason, dysbiosis has been implicated in the onset of several chronic autoimmune or inflammatory pathologies, including IBD (e.g., ulcerative colitis (UC) and Crohn’s disease (CD)), metabolic diseases (e.g., obesity, type diabetes 2 and nonalcoholic fatty liver disease), autoimmune diseases (e.g., rheumatoid arthritis, allergies, and systemic lupus erythematosus), and other disorders (e.g., food intolerances and even colorectal cancer) [1,3]; to date, only few studies linked dysbiosis to primary or secondary immunological oral mucosal disorders [4,5,6]. 

The onset of extra-intestinal pathologies linked with dysbiosis is due to bacterial signals that affect the innate and adaptive immune system. These signals also involve type 3 innate lymphoid cells, which contribute to the differentiation of T and B cells and induce the production of Th17 cells through secretion of IL-22. Moreover, it has been shown that the intestinal microbiota influences the accumulation of IgA-producing cells in the lumen, and that IgA diversity in the intestine is related to changes in microbiota composition. The intestinal microbiota also promotes the differentiation of naive CD4^+^ T cells into Th17, which act at the epithelial level to improve the integrity of the intestinal mucosal barrier. This suggests that a disruption at this level can cause changes in the intestinal barrier and the onset of various pathologies [7] affecting not only the bowel (e.g., UC), but also other organs (i.e., arthritis, uveitis, etc.).

### 1.1. Oral Manifestations in IBD

Apart from the main symptoms related to the gastrointestinal involvement typical of IBD, these patients may present a broad spectrum of non-intestinal signs and symptoms known as extraintestinal manifestations (EIMs): joints, skin, eyes, the biliary tract and the oral mucosa are the most common sites involved [8].

It is estimated that approximately one third of IBD patients may develop EIMs [9]; in particular, oral lesions may anticipate or accompany gastrointestinal illness: patients with IBD may present these manifestations years before the appearance of intestinal symptoms (5–10%), but EIMs are most commonly diagnosed after intestinal involvement has occurred [10]. EIMs can sometimes be confused with other extraintestinal complications related to IBD due to malnutrition, chronic inflammation or side effects of drugs used to treat the disease itself [11].

The etiopathogenesis, classification, and natural history of muco-cutaneous disorders related to IBD have not yet been well defined: in general, oral lesions are found more often in patients with CD compared to UC, in children compared to adults and in males compared to females [8]. 

Differences may be found in the presentation of oral lesions between CD and UC: CD is characterized by both specific and nonspecific oral lesions, while only nonspecific ones are found in UC. In CD, oral lesions are defined as specific if the histopathologic data shows evidence of granulomas (similar to those observed endoscopically in the intestine); these are less common than nonspecific lesions (showed in Table 1) and they include: indurated tag-like lesions, cobblestoning, mucogingivitis, lip swelling, deep linear ulcerations and midline lip fissuring (Table 2).

In particular, cobblestoning and tag-like lesions are considered pathognomonic for CD, but these are generally not associated with active intestinal disease [8]. Nonspecific oral lesions are found more often than the specific ones, and are usually associated with CD and UC; these include recurrent aphthae, angular cheilitis, pustular ulcerations, pyostomatitis vegetans, glossitis, lichen planus and nonspecific gingivitis [12].

Among the nonspecific findings, recurrent aphthae are the most known oral lesions associated with IBD; when the onset of aphthae is associated with systemic disorders, the term ALU is now considered preferable over the previously used RAS (recurrent aphthous stomatitis), since aphtae are considered a secondary manifestation and the different clinical courses of the two conditions require different management strategies [13]. ALU are reported to occur in up to 10% of UC and 25% of CD patients, and they may become more severe in active disease; however, their presence or absence does not correlate with disease activity [14]. Clinically, ALU are shallow, round or oval shaped lesions, granular on palpation; they are often painful, leading to negative effects on patients’ daily activities [15]. Their onset is usually sudden and may be concurrent with a flare-up of intestinal symptoms, or appear simultaneously with other EIMs [16]. Data from literature are conflicting in terms of the association between ALU and the pathological activity: some studies link the presence of oral lesions with the concomitant presence of intestinal symptoms, while other contradicting findings report no statistically significant difference [17,18].

Deficiencies caused by diet or poor absorption of an essential nutrient can cause anaemia and mineral and vitamin deficiency; in particular, vitamins B1, B2, B6, B12, iron, serum ferritin and folic acid deficiencies have been reported in the pathogenesis of oral ulcers, implicating their role in weakening the immune system [19]. Anaemia may also arise from chronic intestinal bleeding associated with iron deficiency, causing angular cheilitis and painful depapillation of the tongue; constant iron and zinc deficiencies may also be linked to erosive and crusty lesions on the lip commissures and perioral region [20].

Regarding IBD therapy, all of the currently used drug classes have been linked to alterations in the oral cavity due to their direct toxic effect on tissues and their indirect immunosuppressive effects [21].

### 1.2. Morphology: Comparative Microscopic Anatomy of Oral and Intestinal Mucosa 

All the organs of the alimentary canal have a common origin in the primitive digestive gut. The oral mucosa, that covers the entire oral cavity, consists of two layers: a stratified squamous epithelium and an underlying connective tissue (lamina propria) that includes blood and lymphatic vessels, as well as nerves and immune cells. However, the oral mucosa varies in structure, function and appearance in different regions of the cavity, and it is divided into lining, masticatory and specialized (gustative) mucosa [22,23], as detailed in Figure 2. Both the lining and masticatory oral mucosae may host aphthous lesions. 

The wall of the small and large bowel is canonically divided into four layers: mucosa, submucosa, muscularis propria and serosa or adventitia. The mucosa is composed of epithelium, lamina propria and muscularis mucosae. A simple columnar epithelium covers the small and large bowel (Figure 2), although many regional differences are present [24,25,26]. 

The similarities between the oral and the intestinal mucosa include: (1) the presence of tight junctions between epithelial cells; and (2) the presence of a basement membrane between the epithelium and the lamina propria.

The equilibrium between the epithelium and the lamina propria is very important for mucosal homeostasis, and both alterations in tight junctions and changes in basement membrane may result in dysfunction of the mucosal barrier, as seen in inflammatory bowel disease [27].

### 1.3. The Intestinal Microbiota and the Surrounding Mucous: The Fifth Layer of the Bowel Wall

As stated before, the wall of the bowel is canonically divided into four layers by morphologists through the observation of histological sections after processing them with reagents, including alcohols that remove mucus and other alcohol-soluble substances. However, in living subjects, the mucosal layer is characterized by the presence of a mix of symbiotic and pathogenic bacteria embedded in the mucus, produced by the epithelial cells. In this mucous matrix, apart from bacteria, are present a number of soluble substances and nanovesicles (i.e., exosomes, microvesicles and outer membrane vesicles), produced by both human cells and bacteria, that actively participate in the regulation of the homeostasis of the intestinal mucosa and, consequently, through lymphatic and hematic circulation, of virtually all of the organs [28,29,30]. Therefore, as already proposed [22], this mucus-microbiotic layer can be considered the real innermost layer of the intestinal wall. The relevance of this hypothesis lies in the fact that the understanding of the pathogenesis of human diseases derives from a precise knowledge of normal morphology, since many (if not all) pathologies derive from an alteration in cell differentiation that, in turn, generates tissutal changes and loss of organ function; thus most (if not all) treatment strategies should aim—when possible—to restore the normal morphology of the organs. The aim of the present paper is to present a novel pathobiological hypothesis and, consequently, a non-invasive therapeutic method.

## 2. Pathobiology of ALU in IBD and Therapeutic Proposal

### 2.1. Pathobiology: A Focus on Dysbiosis

The persistence of dysbiosis causes a state of chronic inflammation linked to the activation of MALT and the release of inflammation mediators. This causes an onset of pathologies even in areas that are physically far away from the gut [1,2,3].

Intestinal epithelial cells represent the main communication barrier between the host environment and the microbiota, and also regulate the impact of the microbiota on the host immune function (Figure 3). 

For example, a healthy intestinal microbial flora promotes regulatory B (Breg) cell differentiation and IL-1b and IL-6 production, and controls inflammatory processes through Breg cells and IL-10 secretion. Gut microbiota influences not only local, but also systemic immunity by bacterial metabolites (such as ligands of aryl hydrocarbon receptor and polyamines) and bacterial components, such as polysaccharide A, that exhibit immunomodulatory action [31]; moreover, studies carried out on animal models have shown that intestinal dysbiosis may be related to the development of autoimmune diseases.

Mechanisms with which intestinal dysbiosis could generate autoimmune activation are not yet well understood. It is thought that they may be related to: -Alteration of Treg/Th17 due to dysregulated TLRs on antigen-presenting cells [27];-Resistance to colonization, i.e., ability of the gut microbiota to limit the proliferation of external pathogens. It has been observed that in patients with autoimmune disorders (e.g., systemic lupus erythematosus), resistance to colonization was lower than in healthy controls;-Superantigens, derived from bacteria and viruses that have the ability to activate immune cells by simultaneously binding to the major proteins of the class II histocompatibility complex (MHC II) present in the antigen-presenting cells and to specific receptors present in activated T cells [32];-Alteration of host antigens and overproduction of autoantigens; in particular, microbiota induces modification of host proteins and creation of neoantigens [27,33];-Mucosal responses to microbiota, i.e., inflammatory cytokines that activate nearby autoreactive cells [30];-Molecular mimicry, i.e., cross-reactive antibody that recognizes shared epitopes of microbial and host tissue proteins, and activation of autoreactive T and B cells [29,34,35].

Thus, the hypothesis we formulate is that ALU is the result of the concomitance of intestinal dysbiosis (and consequent activation of the immune system) and other events, e.g., the microtraumas occurring (frequently and for various causes) in the oral mucosa. Microtraumas can be considered as a stress factor for oral mucosa that induce overexpression, trafficking and surface mislocalization of intracellular proteins that may work, pathogenetically, as autoantigens. Heat shock proteins are an example of intracellular proteins that—after cell stress—may be mislocalized to cell surface by post-translational modifications that trigger unusual intracellular trafficking pathways; in addition, bacterial Hsp60 homologous, i.e., GroEL, can induce the formation of antibodies against it that can also cross-react against surface-exposed Hsp60, generating an autoimmune response by a molecular mimicry mechanism [36,37,38,39].

### 2.2. ALU Treatment: Can Probiotics Be Useful?

In view of the above hypothesis, we suggest that an *ex adiuvantibus* therapy with probiotics could be able to modify the natural course of ALU (Figure 4). 

We cannot yet precisely answer the question “how do probiotics work?” but some theories can be formulated. There are strong functional similarities between the gut and oral biofilms: it is reasonable to speculate that corresponding health-promoting events may occur in the oral cavity to those already reported in the gut. The oral cavity is a large reservoir of bacteria of >700 species and it is closely related to host health and disease [40,41]. In a recent study, it is demonstrated an association between dysbiosis of the salivary microbiota and IBD patients; it was observed that the salivary microbiota in IBD patients significantly differed from that of healthy ones, and found particular bacterial species associated with dysbiosis (*Prevotella* and *Veillonella* were significantly higher in both the CD and UC groups while *Streptococcus*, *Haemophilus, Neisseria* and *Gemella* were significantly lower compared with the healthy ones). It was also showed that the dysbiosis is strongly associated with elevated inflammatory response of several cytokines with depleted lysozyme in the saliva of IBD patients [42]. In the oral cavity, probiotics create a biofilm which matches with carcinogenic bacteria and periodontal pathogens modulating host immune response by strengthening the immune system [43]. There are local (direct) as well as systemic (indirect) events that occur by regulation of the immune response. The potential pathways could include [44]:-Co-aggregation and growth inhibition;-Bacteriocin and hydrogen peroxide production;-Competitive exclusion through antagonistic activities on adhesion and nutrition;-Immunomodulation.

There is an increasing body of evidence suggesting that perturbations of mucosal microbiota can modulate innate and adaptive immune responses, with inflammation arising upon reduction of the number of symbiont microorganisms and/or increase in the number of pathobiont microorganisms (commensal bacteria with pathogenic potential) [45].

Several immune mechanisms, implicated in the remission of ALU, by symbiont bacteria have been hypothesized, including induction of IL-10, suppression of TNF-α and IL-8, and modulation of Toll-like receptors [46].

This hypothesis has been reinforced by some studies that correlate the administration of probiotics to the improvement of autoimmune diseases. For example, it has been observed that in patients with rheumatoid arthritis, the administration of *Lactobacillus casei* increased the serum levels of IL-10 anti-inflammatory cytokine and decreased the levels of proinflammatory cytokines such as TNF-alpha, IL-6 and IL-12 [27].

## 3. Conclusions

Intestinal dysbiosis causes a chronic inflammatory state and activation of the MALT in the gut, which leads to the onset of extraintestinal pathologies [1,2,3]. We hypothesized that ALU could also be caused by intestinal dysbiosis, due to the immunological mechanisms involved in the pathogenesis of the disease [27] and the fact that there are several immune mechanisms implicated in the remission of ALU mediated by symbiont bacteria [36]. By comparing what happens in the intestine [47], we hypothesize that the administration of probiotics can increase the expression of tight junction protein ZO-1, both in terms of transcriptome and protein synthesis, with an improve intestinal barrier function. In fact, it was shown as a result of a chronic inflammatory state levels of TNF-α, IFN-γ and IL-23 stimulate the epithelial barrier breakdown, affecting in particular the expression of proteins forming the tight junctions. This increase can be countered by administering specific probiotic strains including *Lactobacillus salivarius*, *Bifidobacterium lactis*, *Lactobacillus Plantarum* and *Lactobacillus fermentum* [48,49]. In addition to antagonistic act on proinflammatory cytokines, microbial metabolites directly promote the synthesis of the aforementioned tight junction proteins through activation of aryl hydrocarbon receptor, with subsequent activation of nuclear factor erythroid 2–related factor 2 (Nrf2) which has as a final result just the increased synthesis of ZO-1 with consequent strengthening of the epithelial barrier [50].

Therefore, we proposed the use of probiotics as direct therapy for intestinal dysbiosis and as an indirect one for ALU. In particular, preliminary observations lead us to suggest that the therapy with probiotics should be started when the patient first starts to experience the ALU symptoms. Our hypothesis is that this therapy can reduce the duration of the disease by up to three days, through limiting the development of the lesion and favouring the re-epithelization of the lesioned oral cavity mucosa thanks to the molecular mechanisms discussed above (Figure 5).

Experimental and clinical evidence on the use of probiotics for the treatment of oral aphthae are currently very limited, and the etiology and pathogenesis of ALU is currently unknown. So, we think that it would be opportune to carry out in depth studies of this phenomenon, taking into account that host genetics, nutritional deficiencies, and a number of systemic conditions have been recognized as systemic modulating factors of ALU [34,51]. Further studies are needed to establish which immunological mechanisms can be implicated in ALU pathogenesis and modulated by the administration of probiotics.

## Figures and Tables

**Figure 1 ijms-20-05026-f001:**
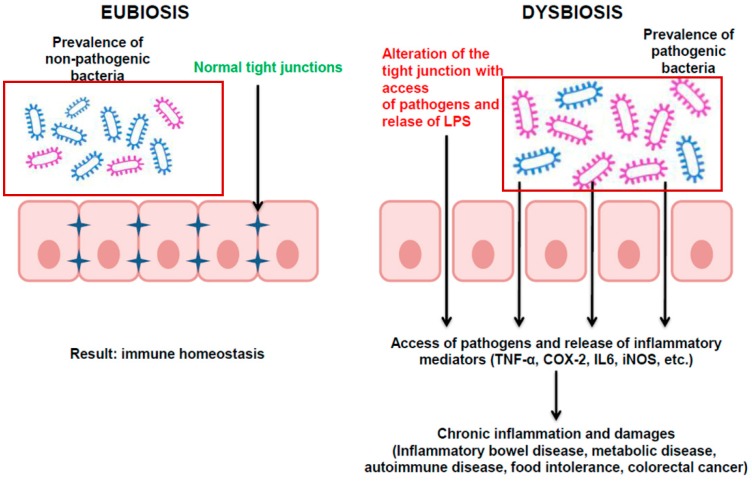
The delicate equilibrium between eubiosis and dysbiosis in the bowels. Eubiosis is the condition in which saprophytic bacteria are present in the mucus-microbiotic layer of the bowel (either the small or the large one). Dysbiosis is a condition in which pathogenic bacteria (Pathogenic bacteria are represented with purple frame, non-pathogenic have a blue frame) predominate and cause changes in the intercellular tight junctions and, in turn, activation of the MALT, leading to tissue damage.

**Figure 2 ijms-20-05026-f002:**
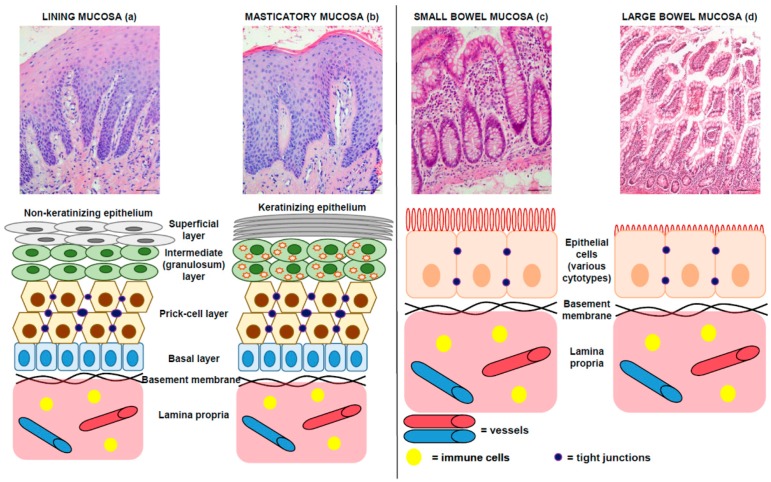
Comparison between oral (both lining and masticatory, respectively **a**,**b**) and bowel (both small and large, respectively **c**,**d**) mucosae. Above: Original pictures, hematoxylin and eosin stainings; bar: 100 micra. Original magnifications: 100×. Below: drawings summarizing the main characteristics of these tissues. In detail, the epithelium of oral mucosa is a stratified squamous epithelium, non-keratinized in the lining mucosa (**a**) and keratinized in the masticatory mucosa (**b**). It is divided into four layers: basal layer, prickle-cell layer, intermediate layer and superficial layer for lining mucosa; and basal layer, prickle-cell layer, granular layer and superficial (keratinizing) layer for masticatory mucosa. In both epithelia, the basal layer consists of cuboidal or columnar keratinocytes that are capable of division so as to maintain a constant epithelial population. Cells arising by division in the basal layers of the epithelium undergo a process of maturation as they are passively displaced toward the surface. In the non-keratinized squamous epithelium, the cytoplasm of intermediate cells does not contain keratin filaments. In keratinizing epithelium, the granulosum stratum is prominent and cells contain intracytoplasmic granules of keratohyaline. The epithelium of small bowel (**c**) covers the intestinal villi and the crypt compartments; it is columnar and composed of various cell types, such as absorptive cells, goblet cells and endocrine cells, in the villi, and stem cells and Paneth’s cells, in the crypts. The epithelium of large bowel (**d**) covers glandular crypts; it is composed of a single layer of columnar cells and consist of absorptive cells that are responsible of water and ion transport, and goblet cells.

**Figure 3 ijms-20-05026-f003:**
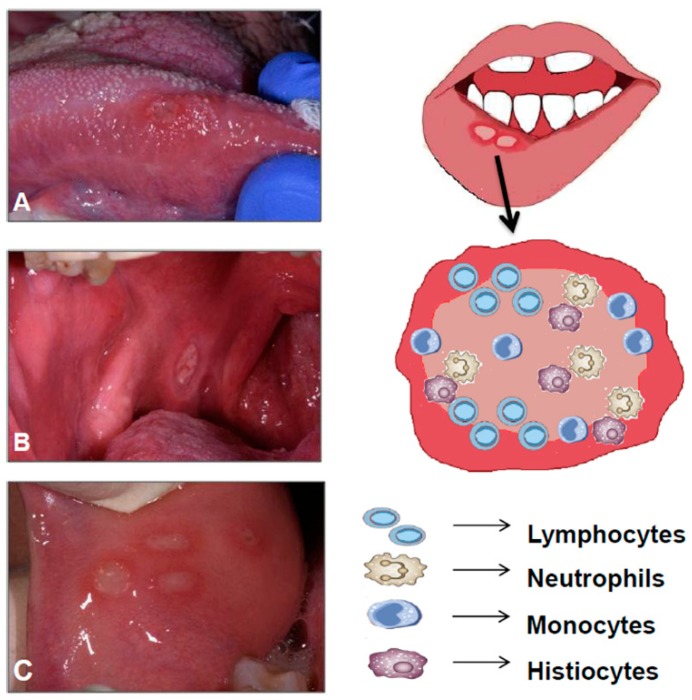
Aphthous-like ulcers: histopathological features. Left: These images show some typical features of ALU: round or oval ulcer covered by a yellow-white fibromembrane with a peripheral erythematous halo; margins may appear indurated and elevated (**A**: ALU in the right side of the tongue; **B**: ALU in the posterior buccal mucosa; **C**: Four concomitant ALU in the anterior buccal mucosa). Right: The ulcerative lesion shows an increased angiogenesis and a mixed inflammatory infiltrate that consists of various leukocytes (lymphocytes, neutrophils, monocytes and histiocytes).

**Figure 4 ijms-20-05026-f004:**
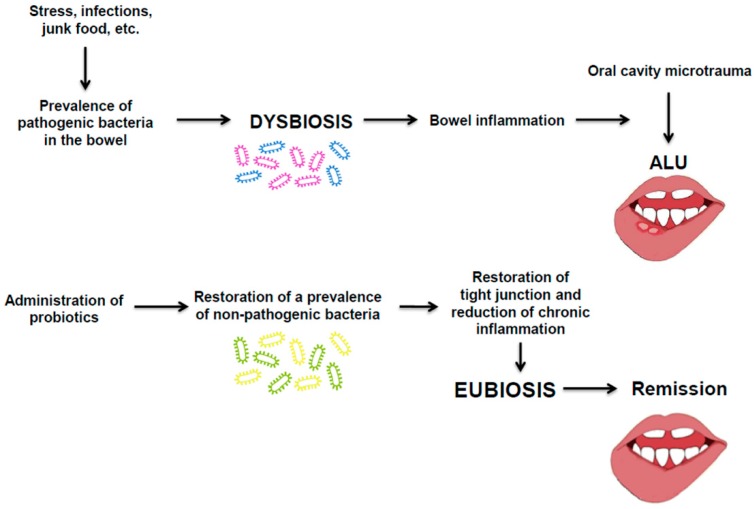
Probiotics as effective therapy for ALU. Restoration of eubiosis can dramatically contribute to remission of ALU by contrasting pathogenic phenomena. (Pathogenic bacteria are represented with purple frame, non-pathogenic with a blue frame; the yellow and green frame indicate the bacterial strains present after administration of probiotics).

**Figure 5 ijms-20-05026-f005:**
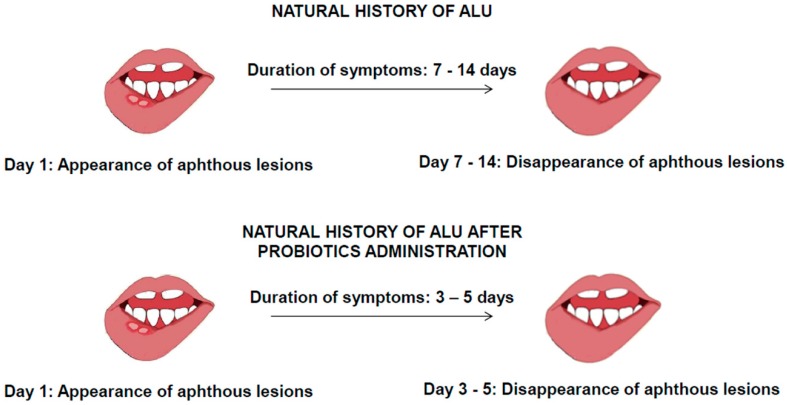
Probiotics as effective therapy for ALU. Hypothesis: Restoration of eubiosis can dramatically contribute to remission of ALU by contrasting pathogenic phenomena.

**Table 1 ijms-20-05026-t001:** Clinical characteristics of nonspecific oral lesions in CD.

Nonspecific Oral Lesions	Clinical Presentation
Aphtous stomatitis	Shallow, round ulcers surrounded by an erythematous halo with a central fibrin membrane
Angular cheilitis	Erythema with/without painful fissures and sores at the corners of the mouth
Glossitis	Painful atrophy of the tongue
Pyostomatitis vegetans	Small exophytic lesions covered with a vulnerable membrane, their cracking and confluence results in the characteristicsign of a “snail track”
Oral Lichen/Oral Lichenoid reactions	Associated to taste disturbances
Gingivitis/Periodontitis	Associated to a vitamin D deficiency

**Table 2 ijms-20-05026-t002:** Clinical characteristics of specific oral lesions in CD.

Specific Oral Lesions	Clinical Presentation
Indurated tag-like lesions (mucosal tags)	White reticular tags (labial and buccal vestibules, retromolar region)
Cobblestoning	Fissured and corrugated swollen mucosa with hyperplastic appearance (posterior buccal mucosa)
Mucogingivitis	Edematous, hyperplastic and granular gingiva (whole gingiva up to the mucogingival line)
Lip swelling	Associated to vertical fissures
Deep linear ulcerations	Associated to hyperplastic margins (vestibule)
Tongue and midline lip fissuring	Lip and tongue fissures

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
