# Peer review of "Probiotics Can Cure Oral Aphthous-Like Ulcers in Inflammatory Bowel Disease Patients: A Review of the Literature and a Working Hypothesis"

_ijms, 2019, doi:10.3390/ijms20205026_

Round 1
Reviewer 1 Report
The present review describes the relation between disbiosis and the onset of different several chronic autoimmune or inflammatory pathologies, metabolic and autoimmune diseases, and other disorders (e.g. food intolerances and even colorectal cancer.
In particular, inflammatory bowel diseases (IBD) can involve areas far away from the gut, including extraintestinal manifestations such us in the oral cavity with the onset of aphthous-like ulcers (ALU). Dysbiosis never has been linked to primary or secondary immunological oral mucosal disorders. On this basis, aim of the paper, is to verify the hypothesis that in IBD patients, ALU are the result of the concomitance of intestinal dysbiosis and other events, such us microtraumas which occurri in the oral mucosa, and that therapy with probiotics can be employed to modify the natural course of ALU. A novel pathobiological hypothesis and a non-invasive therapeutic method based on the use of probiotics has been proposed.
The work is well written and needs only minor revision:
Pag. 3: to facilitate the lecture, another table containing the list of nonspecific oral lesions, should be added.
Pag. 4 line 117: “Table 1” and “Fig. 1” should be changed with “Fig. 2a,b”
Pag. 8 line 237; Lactobacillus casei should be written in Italics.
Author Response
"Please see the attachment."

Reviewer 2 Report
Don’t use abbreviations in the abstract You wrote: “to date, dysbiosis never has been linked to primary or secondary immunological oral mucosal disorders.”
This is not true. Look, for example, at:
Kim YJ, Choi YS, Baek KJ, Yoon SH, Park HK, Choi Y. Mucosal and salivary
microbiota associated with recurrent aphthous stomatitis. BMC Microbiol. 2016 Apr 1;16 Suppl 1:57. doi: 10.1186/s12866-016-0673-z. PubMed PMID: 27036492; PubMed Central PMCID: PMC4818471.
Bankvall M, Sjöberg F, Gale G, Wold A, Jontell M, Östman S. The oral
microbiota of patients with recurrent aphthous stomatitis. J Oral Microbiol. 2014 Oct 29;6:25739. doi: 10.3402/jom.v6.25739. eCollection 2014. PubMed PMID: 25626771; PubMed Central PMCID: PMC4221501.
Hijazi K, Lowe T, Meharg C, Berry SH, Foley J, Hold GL. Mucosal microbiome in patients with recurrent aphthous stomatitis. J Dent Res. 2015 Mar;94(3
Suppl):87S-94S. doi: 10.1177/0022034514565458. Epub 2014 Dec 24. PubMed PMID: 25540188; PubMed Central PMCID: PMC4541092.
You have to cite all of the study about dysbiosis and primary or secondary immunological oral mucosal disorders
“Apart from the main symptoms related to the gastrointestinal involvement typical of IBD, these patients may present a broad spectrum of non-intestinal signs and symptoms known as extraintestinal manifestations (EIMs): joints, skin, eyes, the biliary tract and the oral mucosa are the most common sites involved [5].”
Cite a more general reference like:
The gut and the Inflammatory Bowel Diseases inside-out: the extra-intestinal manifestations. Minerva Gastroenterol Dietol. 2019 Apr 16. doi: 10.23736/S1121-421X.19.02577-7. [Epub ahead of print] PubMed PMID: 30994321.
Regarding figure five you have to clearly define how the 3-5 days come from or you have to delete it. Furthermore, you suppose that with a probiotic therapy the duration of aphthous stomatitis could be shorter: what about a preventive strategy?
Author Response
Response to Reviewer 2 Comments
Point 1: Don’t use abbreviations in the abstract
Response 1: We removed the abbreviation from the abstract
Point 2: You wrote: “to date, dysbiosis never has been linked to primary or secondary immunological oral mucosal disorders.”
This is not true. Look, for example, at:
Kim YJ, Choi YS, Baek KJ, Yoon SH, Park HK, Choi Y. Mucosal and salivary
microbiota associated with recurrent aphthous stomatitis. BMC Microbiol. 2016 Apr 1;16 Suppl 1:57. doi: 10.1186/s12866-016-0673-z. PubMed PMID: 27036492; PubMed Central PMCID: PMC4818471.
Bankvall M, Sjöberg F, Gale G, Wold A, Jontell M, Östman S. The oral
microbiota of patients with recurrent aphthous stomatitis. J Oral Microbiol. 2014 Oct 29;6:25739. doi: 10.3402/jom.v6.25739. eCollection 2014. PubMed PMID: 25626771; PubMed Central PMCID: PMC4221501.
Hijazi K, Lowe T, Meharg C, Berry SH, Foley J, Hold GL. Mucosal microbiome in patients with recurrent aphthous stomatitis. J Dent Res. 2015 Mar;94(3
Suppl):87S-94S. doi: 10.1177/0022034514565458. Epub 2014 Dec 24. PubMed PMID: 25540188; PubMed Central PMCID: PMC4541092.
You have to cite all of the study about dysbiosis and primary or secondary immunological oral mucosal disorders
Response 2: Thanks for the suggestion, we provided to add the indicated reference to our list
Point 3: “Apart from the main symptoms related to the gastrointestinal involvement typical of IBD, these patients may present a broad spectrum of non-intestinal signs and symptoms known as extraintestinal manifestations (EIMs): joints, skin, eyes, the biliary tract and the oral mucosa are the most common sites involved [5].”
Cite a more general reference like:
The gut and the Inflammatory Bowel Diseases inside-out: the extra-intestinal manifestations. Minerva Gastroenterol Dietol. 2019 Apr 16. doi: 10.23736/S1121-421X.19.02577-7. [Epub ahead of print] PubMed PMID: 30994321.
Response 3: We have proceeded to replace the reference
Point 4: Regarding figure five you have to clearly define how the 3-5 days come from or you have to delete it. Furthermore, you suppose that with a probiotic therapy the duration of aphthous stomatitis could be shorter: what about a preventive strategy?
Response 4: We assume a reduction of the duration of the disease (that could be clearly variable) and we added a list of molecular mechanisms that other studies (reference integrated) showed at bowel level. The idea of a preventive strategy is excellent an could be the main topic for a future study.
Extract from the review: By comparing what happens in the intestine [47], we hypothesize that the administration of probiotics can increase the expression of tight junction protein ZO-1, both in terms of transcriptome and protein synthesis, with an improve intestinal barrier function. In fact, it was shown as a result of a chronic inflammatory state levels of TNF-α, IFN-γ and IL-23 stimulate the epithelial barrier breakdown, affecting in particular the expression of proteins forming the tight junctions. This increase can be countered by administering specific probiotic strains including Lactobacillus salivarius, Bifidobacterium lactis, Lactobacillus Plantarum and Lactobacillus fermentum. [48-49]. In addition to antagonistic act on proinflammatory cytokines, microbial metabolites directly promote the synthesis of the aforementioned tight junction proteins through activation of aryl hydrocarbon receptor, with subsequent activation of nuclear factor erythroid 2–related factor 2 (Nrf2) which has as a final result just the increased synthesis of ZO-1 with consequent strengthening of the epithelial barrier [50]
Schoultz I, Keita ÅV. Cellular and Molecular Therapeutic Targets in Inflammatory Bowel Disease-Focusing on Intestinal Barrier Function. Cells. 2019;8(2):193. Published 2019 Feb 22. doi:10.3390/cells8020193 Al-Sadi R., Boivin M., Ma T. Mechanism of cytokine modulation of epithelial tight junction barrier. Front. Biosci. (Landmark Ed.) 2009;14:2765–2778. doi: 10.2741/3413. Roselli M., Finamore A., Nuccitelli S., Carnevali P., Brigidi P., Vitali B., Nobili F., Rami R., Garaguso I., Mengheri E. Prevention of TNBS-induced colitis by different Lactobacillus and Bifidobacterium strains is associ- ated with an expansion of gamma delta T and regulatory T cells of intestinal intraepithelial lymphocytes. Inflamm. Bowel. Dis. 2009;15:1526–1536. doi: 10.1002/ibd.20961. Singh R, Chandrashekharappa S, Bodduluri SR, Baby BV, Hegde B, Kotla NG, Hiwale AA, Saiyed T, Patel P, Vijay-Kumar M, Langille MGI, Douglas GM, Cheng X, Rouchka EC, Waigel SJ, Dryden GW, Alatassi H, Zhang HG, Haribabu B, Vemula PK, Jala VR. Enhancement of the gut barrier integrity by a microbial metabolite through the Nrf2 pathway. Nat Commun. 2019 Jan 9;10(1):89. doi: 10.1038/s41467-018-07859-7. PubMed PMID: 30626868; PubMed Central PMCID: PMC6327034.
Reviewer 3 Report
In this paper, Dr. Cappello et al. achieve state-of-the-art covering dysbiosis associated with inflammatory bowel diseases and extraintestinal manifestations involving the oral cavity. They aimed to correlate the alleviation of aphthous-like ulcers (ALU) linked with intestinal dysbiosis and other events occurring in the oral mucosa and the therapy with probiotics.
However, some issues need to be discussed:
the composition of microbiota residing on oral mucosa in comparison with intestinal one in IBD, which in turn modulates immunity and thereby affects disease progression; how is modulated under probiotics therapy the pathogenetic mechanism of oral epithelial barrier injury in IBD – particularly on tight junction protein components and how is modulated under probiotics therapy
Author Response
Response to Reviewer 3 Comments
Point 1: the composition of microbiota residing on oral mucosa in comparison with intestinal one in IBD, which in turn modulates immunity and thereby affects disease progression; how is modulated under probiotics therapy the pathogenetic mechanism of oral epithelial barrier injury in IBD – particularly on tight junction protein components and how is modulated under probiotics therapy
Response 1: We analysed the oral microbiota and studied the molecular pathway that other works discovered on the intestinal part assume that there is the possibility to apply the same mechanisms at the level of the oral cavity. In particular the new sections added are:
The oral cavity is a large reservoir of bacteria of >700 species and it is closely related to host health and disease [40-41]. In a recent study, it is demonstrated an association between dysbiosis of the salivary microbiota and IBD patients; it was observed that the salivary microbiota in IBD patients significantly differed from that of healthy ones, and found particular bacterial species associated with dysbiosis (Prevotella and Veillonella were significantly higher in both the CD and UC groups while Streptococcus, Haemophilus, Neisseria and Gemella were significantly lower compared with the healthy ones). It was also showed that the dysbiosis is strongly associated with elevated inflammatory response of several cytokines with depleted lysozyme in the saliva of IBD patients [42]. In the oral cavity, probiotics create a biofilm which matches with carcinogenic bacteria and periodontal pathogens modulating host immune response by strengthening the immune system [43].
By comparing what happens in the intestine [47], we hypothesize that the administration of probiotics can increase the expression of tight junction protein ZO-1, both in terms of transcriptome and protein synthesis, with an improve intestinal barrier function. In fact, it was shown as a result of a chronic inflammatory state levels of TNF-α, IFN-γ and IL-23 stimulate the epithelial barrier breakdown, affecting in particular the expression of proteins forming the tight junctions. This increase can be countered by administering specific probiotic strains including Lactobacillus salivarius, Bifidobacterium lactis, Lactobacillus Plantarum and Lactobacillus fermentum. [48-49]. In addition to antagonistic act on proinflammatory cytokines, microbial metabolites directly promote the synthesis of the aforementioned tight junction proteins through activation of aryl hydrocarbon receptor, with subsequent activation of nuclear factor erythroid 2–related factor 2 (Nrf2) which has as a final result just the increased synthesis of ZO-1 with consequent strengthening of the epithelial barrier [50]
Curtis M.A., Zenobia C., Darveau R.P. The relationship of the oral microbiotia to periodontal health and disease. Cell Host Microbe. 2011;10:302–6. doi:10.1016/j.chom.2011.09.008. Arweiler NB, Netuschil L. The Oral Microbiota. Adv Exp Med Biol. 2016;902:45-60. doi: 10.1007/978-3-319-31248-4_4. Said HS, Suda W, Nakagome S, Chinen H, Oshima K, Kim S, Kimura R, Iraha A, Ishida H, Fujita J, Mano S, Morita H, Dohi T, Oota H, Hattori M. Dysbiosis of salivary microbiota in inflammatory bowel disease and its association with oral immunological biomarkers. DNA Res. 2014 Feb;21(1):15-25. doi: 10.1093/dnares/dst037. Epub 2013 Sep 7. Caglar E, Kargul B, Tanboga I (2005) Bacteriotherapy and pro- biotics’ role on oral health. Oral Dis 11(3):131–137.
Schoultz I, Keita ÅV. Cellular and Molecular Therapeutic Targets in Inflammatory Bowel Disease-Focusing on Intestinal Barrier Function. Cells. 2019;8(2):193. Published 2019 Feb 22. doi:10.3390/cells8020193 Al-Sadi R., Boivin M., Ma T. Mechanism of cytokine modulation of epithelial tight junction barrier. Front. Biosci. (Landmark Ed.) 2009;14:2765–2778. doi: 10.2741/3413. Roselli M., Finamore A., Nuccitelli S., Carnevali P., Brigidi P., Vitali B., Nobili F., Rami R., Garaguso I., Mengheri E. Prevention of TNBS-induced colitis by different Lactobacillus and Bifidobacterium strains is associ- ated with an expansion of gamma delta T and regulatory T cells of intestinal intraepithelial lymphocytes. Inflamm. Bowel. Dis. 2009;15:1526–1536. doi: 10.1002/ibd.20961. Singh R, Chandrashekharappa S, Bodduluri SR, Baby BV, Hegde B, Kotla NG, Hiwale AA, Saiyed T, Patel P, Vijay-Kumar M, Langille MGI, Douglas GM, Cheng X, Rouchka EC, Waigel SJ, Dryden GW, Alatassi H, Zhang HG, Haribabu B, Vemula PK, Jala VR. Enhancement of the gut barrier integrity by a microbial metabolite through the Nrf2 pathway. Nat Commun. 2019 Jan 9;10(1):89. doi: 10.1038/s41467-018-07859-7. PubMed PMID: 30626868; PubMed Central PMCID: PMC6327034.
Round 2
Reviewer 2 Report
Thank you for your modifications.
I only ask you to remove IBD and ALU terms from the abstract.
Reviewer 3 Report
The requested changes have been made and the manuscript can be accepted for publication in this form.